# The Role of Retinal Pigment Epithelial Cells in Age-Related Macular Degeneration: Phagocytosis and Autophagy

**DOI:** 10.3390/biom13060901

**Published:** 2023-05-29

**Authors:** Zhibo Si, Yajuan Zheng, Jing Zhao

**Affiliations:** Department of Ophthalmology, The Second Hospital of Jilin University, Changchun 130000, China

**Keywords:** age-related macular degeneration, autophagy, oxidative stress, phagocytosis, retinal pigment epithelial cell

## Abstract

Age-related macular degeneration (AMD) causes vision loss in the elderly population. Dry AMD leads to the formation of Drusen, while wet AMD is characterized by cell proliferation and choroidal angiogenesis. The retinal pigment epithelium (RPE) plays a key role in AMD pathogenesis. In particular, helioreceptor renewal depends on outer segment phagocytosis of RPE cells, while RPE autophagy can protect cells from oxidative stress damage. However, when the oxidative stress burden is too high and homeostasis is disturbed, the phagocytosis and autophagy functions of RPE become damaged, leading to AMD development and progression. Hence, characterizing the roles of RPE cell phagocytosis and autophagy in the pathogenesis of AMD can inform the development of potential therapeutic targets to prevent irreversible RPE and photoreceptor cell death, thus protecting against AMD.

## 1. Introduction

Age-related macular degeneration (AMD) is the leading cause of vision loss and irreversible blindness in the elderly worldwide; the number of people with AMD is expected to reach 288 million by 2040 [1]. Clinically, AMD is often classified as either dry or wet. Dry AMD, also known as non-neovascular AMD, is more common than wet AMD, accounting for 90% of total AMD cases. In dry AMD, pathological changes primarily occur in the retinal pigment epithelium (RPE) and Bruch’s membrane. Dry AMD progresses to a late stage of “geographic atrophy” characterized by the destruction and death of the RPE and photoreceptor cells. Chronic damage of the RPE leads to the formation of Drusen [2]. Wet AMD, also known as neovascular AMD, is characterized by choroidal neovascularization and an abnormal increase in vascular endothelial growth factor (VEGF) expression [3]. RPE cells are highly specialized epithelial cells with their apical surface in close contact with the outer segments of photoreceptors (POS) and the basal surface adherent to Bruch’s membrane. These cells play an important role in maintaining daily photoreceptor renewal and in the nutrition and function of the choriocapillaris [4]. Cells in the RPE phagocytose POS to ensure the continuous renewal of photoreceptor cells. At the same time, persistent photooxidative stress leads to the accumulation of reactive oxygen species (ROS). Under normal physiological conditions, the abundant antioxidants in RPE cells scavenge oxygen radicals to maintain homeostasis. In contrast, RPE cells in AMD exhibit increased levels of apoptosis, autophagy, and incomplete POS digestion, which in turn increase the burden of oxidative stress [5]. Persistent oxidative stress impairs the phagocytosis and autophagy of RPE cells, thereby increasing protein aggregation and activation of inflammatory vesicles.

Currently, most treatment options for AMD are administered to patients with advanced wet AMD, whereas no therapeutic agent can effectively prevent irreversible RPE and photoreceptor cell apoptosis and loss [6,7]. Indeed, an increasing body of evidence suggests that abnormal phagocytosis and autophagy of RPE cells contribute to AMD pathogenesis. Accordingly, therapeutics that target RPE cell phagocytosis and autophagy have been proposed as potential treatment options for AMD. However, although selective autophagy has been described in other systemic diseases, the safety and efficacy of various possible autophagy-related targets remain to be discussed in the context of RPE cells [8,9]. In this review, we summarize the roles and mechanisms of RPE cell phagocytosis and autophagy in AMD and discuss the possible therapeutic targets for AMD. As such, this article is expected to provide new strategies for the prevention and treatment of AMD.

## 2. Pathogenesis of AMD

### 2.1. Risk Factors for AMD

Several blood vessels converge at the macula, which is therefore exposed to an abnormally high oxidative stress environment. Additionally, the RPE is a major source of ROS, owing to the abundance of mitochondria and extremely high metabolic activity [10]. The large amount of oxygen radicals that accumulates owing to persistent photooxidation has the potential to cause oxidative damage. Under normal physiological conditions, the antioxidant systems of RPE cells, such as superoxide dismutase, glutathione, and melanosomes, participate in scavenging oxygen radicals.

AMD is associated with several environmental and genetic factors that increase the oxidative stress burden of the RPE [11,12]. Among these, age is a major risk factor for the development of AMD. With increasing age, the number of photoreceptors, ganglion cells, and RPE cells are reduced, and adverse reactions such as melanin loss, Bruch’s membrane thickening, and lipofuscin accumulation occur in RPE cells [13].The accumulation of proteins, lipids, and damaged DNA in aging tissues has a negative impact on the physiological function of cells. Consequently, the antioxidant function of RPE cells is compromised, making it incapable of removing the accumulated cell debris caused by the incomplete degradation of POS, resulting in a progressive increase in ROS. Thus, the increased sensitivity of aging cells to ROS aggravates oxidative stress. In addition, studies have found that age-related decreases in ubiquitin-dependent protein hydrolysis can lead to undegraded proteins forming basement deposits under the retina, which may contribute to the occurrence of AMD [14,15]. Smoking is considered to be the strongest non-genetic risk factor for AMD. Tobacco contains many strong oxidants, such as ROS, epoxides, and nitric oxide. Although the exact role that the oxidants in tobacco play in AMD is unknown, strong oxidants are known to deplete ascorbic acid and protein sulfhydryl groups in tissues, leading to the oxidation of DNA, lipids, and proteins [16,17,18,19]. In addition, obesity and a high-fat diet have recently been found to exacerbate subretinal inflammation and choroidal neovascularization and are the second most important risk factors for advanced AMD after smoking. A possible pathogenic mechanism is that a high-fat diet causes dysregulation of intestinal flora and impaired intestinal barrier function, which in turn leads to the release of microbial particles from the gut into the bloodstream. This induces the production of inflammatory factors and thus promotes chronic inflammation and pathological angiogenesis [20,21,22] (Figure 1).

### 2.2. RPE Cells in the Pathological Changes of AMD

The RPE is a part of the blood–retinal barrier, which consists of a single layer of RPE cells. Healthy RPE cells are hexagonal in shape, with tight connections forming between each RPE cell [23]. RPE cells have two types of microvilli that face the interphotoreceptor matrix and the photoreceptor outer segment [24]. Many organelles with specific functions can be found in RPE cells, including mitochondria, phagosomes, and lysosomes; these are located on the basal side of cells and participate in the renewal of phagocytes at the outer end of photoreceptors [25]. In humans, each RPE cell is responsible for the metabolism and renewal of 30–40 photoreceptor cells. The central fovea of the macula only has cone cells and RPE cells, and cone cells are the densest in the fovea. Consequently, the symptoms of impaired visual sensitivity and dark adaptation are particularly evident in patients with early AMD [26,27]. POS that are phagocytosed daily by RPE cells contain a large amount of unsaturated fatty acids, which are oxidized to produce a large amount of ROS during phagocytosis. As a result, the processes of continuous metabolism and renewal of the POS are the major sources of ROS in RPE cells [28]. The pathological changes that occur in early AMD are the degeneration of RPE cells and mesenchymal morphology, while photoreceptor function remains intact [29,30]. As the disease progresses, excessive metabolic wastes can no longer be degraded by autophagy and the intracellular lysosomal system of the RPE, which eventually leads to cell degeneration [15]. In AMD, the blood-retinal barrier weakens, lipofuscin is deposited, and failed lysosomes enlarge, as tight connections between cells separate, thin, or divide. During normal aging, Bruch’s membrane thickens. Nutrients and oxygen from the choroid and waste from the RPE must pass through Bruch’s membrane to maintain retinal homeostasis [31]. In the early stages of AMD, Bruch’s membrane thickens compared to normal aging. Electron microscopy revealed that abnormal deposits occurred above and below the RPE basement membrane in early and mid AMD, respectively, and may be involved in drusen formation. This process of change in Bruch’s membrane thickening reduces nutrient and substance exchange between RPE and choroid, leading to RPE photoreceptor dysfunction in AMD. As AMD progresses to advanced stages of the disease, Bruch’s membrane becomes thinner [32]. As one of the primary functions of the RPE is to maintain the function of photoreceptors, RPE cell degeneration leads to photoreceptor dysfunction and even death, resulting in AMD-related visual impairment. Therefore, it is speculated that RPE cells may develop AMD-related dysfunction if their phagocytic and autophagic functions are disrupted.

## 3. Mechanism of RPE Cell Phagocytosis and Autophagy

### 3.1. Phagocytosis

Photoreceptors are susceptible to photooxidative damage, and their long-term health depends to a large extent on the processing of the aged portion of their outer segments. RPE cells are specialized for the phagocytosis of POS and are the only means of removing their aged portion, a process also known as heterophagy [33]. Each RPE cell is in close contact with approximately 30 photoreceptor (cone and rod) cells, which they will phagocytose and then remove their outer segments. Notably, the shedding and phagocytosis of POS are regulated by circadian rhythms, with most occurring soon after the onset of light [34].

The tip of POS contains high concentrations of free radicals, photodamaged proteins, and lipids. POS is maintained at a constant length through continuous maintenance of the balance between the shedding of POS tips and the formation of new POS. Exposed phosphatidylserine gradually accumulates at the POS tip under light exposure [35]. Photoreceptor cells are in contact with the RPE in a fixed manner, and the apical microvilli of RPE cells wrap around the POS before phosphatidylserine accumulation [36,37]. Phagocytosis of POS can be divided into four distinct phases: recognition and binding of the POS, POS ingestion, phagosome formation, and phagosome digestion.

In the first step of phagocytosis, a receptor tyrosine kinase (MerTK) mediates the specific binding of POS to the RPE membrane, which in turn activates a second messenger that transduces the signal to the cytosol [38]. Notably, the α_v_β_5_ integrin receptor is essential in the POS binding process. MerTK itself does not bind directly or indirectly to actin but requires the α_v_β_5_ integrin receptor to provide traction for mechanical attachment to actin. The α_v_β_5_ integrin receptor itself is triggered by MerTK [39], and so these two processes interact with each other, mediated by the activation of focal adhesion kinase [40,41,42]. Therefore, cells lacking the MerTK receptor can bind to but not ingest POS [33] (Figure 2). The maturation of phagosomes and their fusion with lysosomes occur during the second half of heterophagy in RPE cells. Cathepsins are proteases that degrade proteins. Currently, the primary role of cathepsin D (CTSD) in RPE cells has been identified as the degradation of POS into glycopeptides in lysosomes. A study found that mice lacking CTSD develop retinal degeneration [43,44]. Additionally, cystatins are inhibitors of lysosomal cysteine proteases highly expressed in RPE cells, and mutations in cystatin genes have been shown to be closely associated with RPE degeneration and AMD pathogenesis [45,46]. A recent study identified βA3/A1-crystallin as a novel lysosomal component in the RPE that regulates phagocytosis and autophagy [47].

It has been found that the phagocytic capacity of RPE in patients with AMD is greatly reduced compared to normal age-matched donors. After the specific binding process, the outer segment of the photoreceptor is fused with the lysosome, effectively recycling vitamin A derivatives and lipids. Failure of the fusion process results in the enlargement of the lysosome, which occurs to a great extent in AMD [48]. The phagocytosis of RPE cells decreases moderately with age, and the autophagy function shows a similar intracellular protein pathway to heterophagy. Therefore, the slowing down of autophagy with age may lead to an associated weakening of phagocytosis [49].

### 3.2. Autophagy

RPE cells are not only one of the most active phagocytic cells in the body but are also extremely metabolically active post-mitotic cells with high rates of autophagy. Mammalian cells often exhibit three types of autophagy: microautophagy, chaperone-mediated autophagy, and macroautophagy. Macroautophagy is the most common autophagy in RPE cells and differs from the other two types in that it depends on the formation of autophagosomes. The contents of autophagosomes are degraded by lysosomal enzymes after fusion with lysosomes, whereas, in microautophagy and chaperone-mediated autophagy, the damaged proteins, lipids, and organelles are directly transported to lysosomes. Impaired autophagy in RPE cells can lead to the accumulation of damaged proteins, which eventually leads to cell degeneration and death.

#### 3.2.1. Autophagy Mechanism

Autophagy is a highly complex intracellular garbage removal pathway, the first step of which includes autophagosome formation. Autophagosomes comprise double-membrane vesicles that participate in the clearance of damaged or nonfunctional organelles, proteins, and other substances [50]. Autophagosome formation requires the participation of a series of autophagy-related genes and proteins, including proteins from the ATG (autophagy-related genes) family and LC3 (microtubule-associated protein 1 light chain 3). Autophagosomes are transported throughout the cell via microtubule dynamics to allow them to fuse with lysosomes. Subsequently, the autophagosome contents are degraded by acid hydrolase, proteases, and other enzymes, allowing them to be recycled. Initiation and activation of the autophagy pathway are highly regulated processes, regulated by myriad signaling pathways and molecules, including mTOR (mammalian target protein of rapamycin), AMPK (adenosine 5‘-monophosphate (AMP)-activated protein kinase), and ULK1/2 (unc-51 like autophagy activating kinase) [51,52] (Figure 3).

#### 3.2.2. Autophagy Pathways

Oxidative stress induces antioxidant responses in RPE cells, which are associated with the activation of autophagy via the p62/Keap1/Nrf2 pathway [53,54,55]. Most anti-oxidation protective mechanisms involve and are mediated by Nrf2, which typically interacts with Keap1 to maintain intracellular redox homeostasis [56,57]. In the absence of oxidative stress, Keap1 degrades Nrf2, thereby inhibiting the signal transduction and regulating transcription to low levels. However, under acute oxidative stress, Keap1 can interact with ROS and undergo a conformational change to both inhibit Nrf2 degradation and release large amounts of Nrf2. Nrf2 translocates to the nucleus and binds to antioxidant response elements in the promoter to initiate transcription and protect cells from oxidative damage [58,59]. The p62 protein was found to selectively sort between the autophagic and proteasomal degradation pathways [60] and to selectively target ubiquitinated proteins for autophagy. Additionally, it regulates intracellular oxidative homeostasis by disrupting the Nrf2–Keap1 complex and interacting with the Nrf2-antioxidant response element pathway [61]. Nrf2 forms a regulatory loop with p62 where Nrf2 activates p62 expression, while p62 promotes the nuclear localization of Nrf2. Oxidative stress has also been found to adversely affect Nrf2 signaling in RPE cells with AMD [62,63].

#### 3.2.3. Autophagy and Oxidative Stress

Oxidative stress and autophagy interact during the pathogenesis of AMD. Oxidative stress is primarily characterized by increased levels of ROS, which are produced in great amounts in RPE and photoreceptor cells, owing to their high metabolic activity and abundant mitochondria [64]. In addition, photooxidative stress induced by both continuous light exposure and the presence of rhodopsin renders the macula a high oxidative stress environment for long periods of time [65,66]. Under normal physiological conditions, antioxidants in the RPE can neutralize ROS. The function of the antioxidant system mainly depends on antioxidant enzymes and DNA repair proteins. With an increase in age, the amount of these two proteins decreases simultaneously, the activity of proteins decreases, and the ability of RPE cells to neutralize ROS is weakened, leading to the deposition of harmful proteins and the formation of lipofuscin. In turn, the accumulation of lipofuscin acts as a positive feedback signal, increasing the sensitivity of RPE cells to light-induced oxidative stress, thereby increasing the level of oxygen radicals, aggravating protein misfolding, and promoting further oxidative stress [14,67,68]. Impaired autophagy in RPE has also been found to promote inflammation. In an in vitro experiment, mouse RPE cells with abnormal autophagy function were co-cultured with bone-marrow-derived macrophages. Inflammatory caspase-1, IL-1β and IL-6 cytokines, nitrite oxides, and pro-angiogenic protein levels were significantly increased after co-culture. This suggests that the activation of the inflammasome occurs after autophagy defects in RPE [69].

It has been found that oxidative stress injury in vitro can activate the proliferation and dedifferentiation of Muller cells to prevent apoptosis of retinal neurons. It was also speculated that glial cells could not only keep neurons alive, but could also activate their cell cycles, ultimately preserving retinal function [70,71]. The reactivity of oxidative stress has been reported in microglia cells. However, when the stress is permanent, the sustained inflammatory response may become dysfunctional, changing the integrity of the retina and even causing the death of neurons, leading to retinal degeneration and disease deterioration [72,73]. Accumulation of advanced glycation end products (AGEs) and enhanced activation of AGE receptors (RAGE) were observed in oxidative stress environments. RAGE are widely found in microglia, macrophages, and monocytes, and are highly expressed in RPE cells, photoreceptors, and chorionic membranes in advanced AMD [74]. RAGE activation promotes CNV by regulating angiogenic activity, activating immune cells, and upregulating proinflammatory factors [75,76,77].

Autophagy exhibits a biphasic effect in AMD: in early stages, the level of autophagy increases to compensate for organelles damaged by oxidative stress, whereas in more advanced stages, autophagy appears to be decreased [78]. Importantly, the level of autophagy in RPE cells also notably declines with aging.

## 4. Roles of RPE Cell Phagocytosis and Autophagy in the Development of AMD

### 4.1. Aging

Senescent mitochondria produce more ROS than young mitochondria. Lipofuscin accumulation is a possible explanation for the decrease in autophagic activity with aging [79]. The initial stage at which lipofuscin appears is currently believed to be when the accumulation of oxidized LDL and lipid peroxidation of intracellular metabolites occurs [80]. Lipofuscin has been found to irreversibly inhibit lysosomal function, affecting autophagy [81,82,83], and once formed, it cannot be degraded by the proteasome or lysosomal enzymes or transported to the extracellular space by extracellular activity [84]. In patients with AMD, lipofuscin is redistributed within the RPE, sometimes forming large deposits or particles within the cell. These deposits can lead to cell dysfunction. This is because visible light irradiation of lipofuscin can cause lipid peroxidation, and the production of hydrogen peroxide can damage mitochondrial DNA in RPE cells. In addition, the risk of chronic oxidative stress damage of RPE increases with age due to the accumulation of photophysin [85,86]. Lipofuscin itself has been found to be toxic and able to affect proton pump activity on the lysosomal membrane, increase lysosomal pH, and impair lysosomal function. With increasing lipofuscin accumulation, lysosomal function is further impaired, which may lead to defective mitochondrial function and further aggravation of oxidative stress [80,87].

### 4.2. Lysosomal Dysfunction

The strong phagocytic and autophagic activity of RPE heavily depends on normal lysosomal function. If lysosomal proteins responsible for the maintenance of retinal homeostasis become dysfunctional, the ability of RPE to remove cellular waste and maintain homeostasis can be significantly affected. Under normal conditions, protein misfolding induced by oxidative stress can be repaired by heat shock proteins. If the function of these misfolded proteins is insufficient, individual polypeptides are ubiquitinated and targeted to the proteasome for degradation, while aggregated polypeptides are degraded by autophagy [68,88]. Mammalian target protein of rapamycin (mTOR) has been shown to be significant in lysosomal-mediated autophagy pathways and is highly conserved in evolution. It plays a central role in cell growth, cell survival, autophagy, and metabolism through two distinct protein complexes, mTOR complex 1 (mTORC1) and mTOR complex 2 (mTORC2), such as that expressed in RPE cells [89]. Activation of mTORC1 can impair lysosome function and inhibit autophagy [90]. In animal models of AMD, decreased phagocytosis of RPE was observed. This is attributed to increased iron levels because the mRNA and protein expression of several iron-regulating molecules increases significantly with age, leading to the accumulation of excess iron, which is toxic to RPE cells and impairs phagocytosis and lysosome function [91]. During AMD, mTOR activation is increased in senescent RPE cells; therefore, to induce autophagy, the inhibition of mTORC1 activity is essential [92]. Proton pumps located in the lysosomal membrane play a key role in the maintenance of acidic microenvironment of the lysosomes. It has been recently reported that βA3/A1-crystallin modulates the activity of the lysosomal proton pump via the AKT/mTOR pathway, thus regulating lysosomal degradation in the RPE [93,94]. For this reason, βA3/A1-crystallin could be a potential therapeutic target for the treatment of AMD.

### 4.3. Mitochondrial Dysfunction

Mitochondria are the primary source of ROS. Daily phagocytosis by RPE cells makes mitochondria susceptible to changes in redox status, thereby increasing mtDNA damage. Mitochondria exert autophagy to selectively remove defective mitochondria with impaired oxidation capacity and decreased integrity [95,96]. A decrease in mitochondrial autophagy was observed with age, which led to the accumulation of damaged mitochondria and an increase in oxidative stress and apoptosis [97]. There is evidence that mtDNA damage is common during aging, and this may explain the changes in mitochondrial structure and gene mutations that occur during normal aging. Additionally, this damage is more widespread during the pathogenesis of AMD, with the extent of mtDNA damage correlating with AMD severity [98]. Mitochondrial changes in AMD have been confirmed in several studies, with increased mtDNA damage in RPE cells of patients with AMD compared to controls, and significantly more in the macula than in the periphery [99,100]. Compared with normal RPEs, RPEs in AMD showed increased sensitivity to oxidative stress, produced higher ROS levels under stress conditions, and decreased mitochondrial activity and ATP production. Terluk found that mtDNA damage in AMD is located in the region encoding electron transport chain genes, resulting in impaired ATP production. In turn, insufficient cellular energy production causes an imbalance between apoptotic and antiapoptotic signaling, further causing dysfunction and necrosis of RPE cells [101,102]. Intracellular mtDNA induces the secretion of inflammatory cytokines IL-6 and IL-8, as well as activating the NLRP3 inflammasome in RPE cells, which are closely related to the onset and progression of AMD [103,104]. In addition, the number of mitochondria has been found to be significantly reduced in patients with AMD compared to age-matched controls [105]. Decreased levels of a mitochondrial heat shock protein (mtHsp70), which functions as a molecular chaperone for the repair of misfolded proteins, has also been observed in AMD; therefore, it is speculated that decreased mtHsp70 levels may have an impact on the mitochondrial function and limit energy production [106]. In a proteomic study, it was found that the expression of ATP synthase subunit in RPE was reduced in patients with AMD. The ATP synthase complex is involved in maintaining mitochondrial morphology and mitochondrial membrane potential, and reduced expression of ATP synthase subunits in AMD may lead to defects in key mitochondrial functions [99,107]. Increased PAPR2 expression, decreased AMPK activity, and overactivation of the mTOR pathway were also observed in AMD RPE cells compared to normal RPEs. Metabolomics and lipidomics also show dysregulation of glycerol phospholipid metabolism, which is involved in autophagy, compared with normal RPE cells. All these suggest that metabolic pathway disorders play an important role in the occurrence and development of AMD [108,109].

Mitochondria can interact with phagocytes to initiate selective autophagy with the help of selective autophagy receptors [110]. Damaged mitochondrial fragments can undergo mitochondrial ubiquitination and lysosomal degradation by autophagic clearance through the Pink1-Parkin-mediated signaling pathway [111]. Mitochondrial autophagy protects mitochondria from environmental damage, such as chronic oxidative stress, and the absence of mitochondrial autophagy can lead to elevated ROS levels in any circumstance. Pharmacologically inhibiting mitochondrial fission may improve phagocytosis of POS and maintain normal mitochondrial fusion function, providing a novel therapeutic target for AMD [112].

### 4.4. Loss of Oxidative Stress Homeostasis

As previously mentioned, the macula is chronically exposed to a high-risk oxidative stress environment. Under normal physiological conditions, oxidative and antioxidant systems are in dynamic equilibrium. If for any reason the oxidative system is strengthened or the antioxidant system is attenuated, then the balance is disrupted and RPE cells experience pathological damage. RPE cells constitute a main source of oxidative stress, owing to their unique function in POS phagocytosis, and the ROS generated during this process aggravate the burden of oxidative stress. In the pathogenesis of AMD, RPE dysfunction caused by oxidative stress can lead to incomplete intracellular digestion of POS, which in turn exacerbates the accumulation of cellular metabolic waste products, including lipofuscin [113]. The three basic elements of the antioxidant system are antioxidant enzymes, DNA repair proteins, and small molecules of antioxidants, but the latter are much less important than the first two. Levels of antioxidant proteins and small molecules of antioxidants decline with age, in addition to the ability of mitochondria and nuclei to repair DNA [114,115,116]. Antioxidant enzymes and DNA repair proteins are the main components of the antioxidant function of cells. The age-related decline in antioxidant system efficacy is closely related to the age-related downregulation and decline in the activity of major proteins. These include NFE2-like BZIP transcription factor 2 (NRF2, NFE2L2), as well as nuclear and mitochondrial superoxide dismutase (SOD1 and SOD2, respectively) [66,117].

Oxidative stress reflects the unbalanced production of reactive oxygen species and antioxidant capacity in cells. Due to its energy-intensive state and constant exposure to light, the retina is vulnerable to ROS, which, as a by-product of retinal cell generation, is a signal sensor involved in the PI3K/AKT/mTOR signaling pathway [118,119]. Increased intracellular ROS levels activate kinases, including MARK and PI3K, and lead to the amplification of their downstream signals [120]. RPE autophagy and oxidative stress have an interactive relationship: when autophagy is upregulated through mTOR, the production of ROS is decreased, and thus defective autophagy can aggravate oxidative stress in AMD. Some researchers have in fact observed differences in autophagic flux by creating models of acute and chronic AMD. They found that acute oxidative stress stimulates an increase in autophagy, whereas chronic oxidative stress decreases it [78]. This suggests that autophagy may play a dual role in AMD; inhibition of autophagy may protect photoreceptor cells from light-induced damage, while excessive autophagy may be detrimental to the protective effects of RPE cells.

## 5. Potential Therapeutic Targets for AMD

### 5.1. Phagocytosis Targets

The RPE plays an important role in maintaining visual function and visual cycling by exerting its phagocytic function. The visual cycle involves a series of biochemical reactions, including regeneration of visual pigment clusters, the 11-cis-retina, and removing its toxic byproducts from the retina, supporting visual function and the survival of retinal neurons [33,121,122]. The *RPE65* gene encodes the trans retinol acetate isomerase RPE65, which is a key visual cycle enzyme that exists primarily in the endoplasmic reticulum of RPE cells and participates in the catalytic isomerization of retinol. All-trans retinol acetate can be transferred to 11-cis retinol, which is critical to the visual cycle [123,124]. Retinaldehyde is a key intermediate in visual desensitization and resensitization that binds to opsin, forming visual pigments. However, during the visual cycle, when 11-cis retina regenerates in the retina, toxic by-products such as all-trans retina and N-retinal-N-retinolamine (A2E) are produced, which are thought to be related to the development of non-exudative AMD. DNA damage, oxidative stress, and mitochondrial dysfunction are associated with the abnormal accumulation of N-retinylidene-N-retinylethanolamin(A2E) in the retina and RPE [125].

Several therapeutic approaches targeting RPE65 have been studied and developed, including gene therapy and small-molecule drug therapy. Gene therapy primarily introduces the normal *RPE65* gene into the retinal pigment epithelial cells of patients through gene transfection or carrier transfection. Consequently, the expression of RPE65 protein is increased, thus promoting retinal circulation metabolism and reducing the accumulation of toxic products [126]. Meanwhile, small-molecule drug therapy primarily refers to visual cycle inhibitors, which inhibit RPE65 activity, reduce the formation and accumulation of retinaldehyde, and regulate the excessive activity of the visual cycle, thus improving AMD and other diseases related to the dysfunction of retinal pigment epithelial cells [127].

CD36 is a transmembrane glycolipoprotein that has an important role in RPE cells, particularly in phagocytosis of the outer segment of RPE. CD36 is expressed on the medial cilia of the RPE and can mediate the recognition and uptake of fatty acids and oxidized low-density lipoprotein (oxLDL) in the external environment. However, CD36 expression levels in RPE cells are influenced by external environmental stimulation. For example, high levels of oxLDL can induce autophagy in the RPE by activating CD36 expression and increasing ROS levels [128,129]. In addition, CD36 can induce activation of the NLRP3 inflammasome, leading to the release of pro-inflammatory factors and further promoting the autophagy process of RPE [130]. CD36 is also involved in the process of heterophagy in RPE. That is, CD36 promotes the phagocytosis of apoptotic cells and photoreceptors by mediating the recognition and binding of phosphatidylserine (PS) by RPE in the external environment [131]. CD36 also regulates RPE heterophagy by regulating intracellular calcium levels and cytoskeletal recombination [132].

### 5.2. Autophagy Targets

Modulation of oxidative stress and the delaying of AMD progression by adjusting RPE cell autophagic activity are being recognized as novel therapeutic directions for AMD. Enhancing autophagy in dry AMD can alleviate the degenerative changes in RPE cells, while anti-angiogenic effects can be enhanced by inhibiting autophagy in wet AMD. However, as previously mentioned, the biphasic effect of autophagy in AMD and the different phenotypes of AMD make the development of autophagy-targeted therapies both complex and challenging.

Selective targeting of the mTOR/autophagy pathway could become a new paradigm for the treatment of AMD. The mTOR pathway is an important regulator of cell metabolism and growth and a key molecule in the regulation of autophagy [133]. mTOR is a protein kinase that regulates biological processes such as metabolism, proliferation, differentiation, and autophagy by phosphorylating downstream target proteins. In RPE cells, the mTOR pathway is involved in the negative regulation of autophagy [134]. Specifically, mTOR inhibits autophagy through two mechanisms: (1) phosphorylating and activating negative regulators of ULK1 (unc-51-like kinase 1), preventing ULK1 from forming complexes with autophagy-related proteins such as ATG13 and FIP200, thereby inhibiting autophagosome formation; (2) phosphorylating and inhibiting the phosphorylation of ATG13, further inhibiting autophagosome formation and autophagy [135,136].

Rapamycin has been reported to inhibit mTORC1 and activate autophagy, preserving the photoreceptor function by blocking the deleterious degeneration of RPE cells in response to oxidative stress [137]. Mitogen-activated protein kinase (MAPK) and pro-inflammatory cytokine TNFα promote mTORC1 activation by inhibiting TSC1/TSC2 [138]. Another study demonstrated that rapamycin interferes with VEGF-A function, reduces endothelial cell proliferation, and inhibits choroidal angiogenesis [139]. However, given that the activation of autophagy and interference with VEGF function have opposite outcomes in wet AMD, further studies are needed to determine the potential for rapamycin in AMD treatment.

The AMP-activated protein kinase (AMPK) pathway is closely related to autophagy regulation. As a protein kinase, AMPK is activated by the cellular AMP/ATP ratio. In RPE cells, the AMPK pathway is involved in the positive regulation of autophagy [140,141]. On the one hand, AMPK can enhance the binding of ULK1 with other autophagy-related proteins by directly phosphorylating ULK1, promoting the formation of autophagosomes. On the other hand, AMPK activation can relieve mTORC1’s inhibition of autophagy by inhibiting its activation and promote ULK1 phosphorylation, thus further promoting the formation of autophagosomes [142]. In addition, AMPK activation can promote autophagy by encouraging ATG9 migration and enhancing the activity of acidic proteases. Indeed, AICAR, an activator of the AMPK pathway, protects RPE cells from oxidative stress by inducing autophagy [143,144].

Long non-coding RNAs (lncRNAs) are an important class of transcriptional products involved in various pathological and physiological regulatory processes. Some lncRNAs can affect autophagy by regulating mTOR activity. For example, the lncRNA GAS5 can inhibit mTOR activity and promote autophagy. GAS5 is downregulated in tumor cells; however, its overexpression can inhibit tumor cell proliferation and migration by inactivating the mTOR pathway [145,146]. In addition, treatment of RPE cells with the mTOR/AMPK pathway activator Decorin (DCN) significantly reduces apoptosis and ROS levels, protecting RPE cells from oxidative stress and apoptosis through the promotion of autophagy. Hence, DCN represents a potential therapeutic target for AMD [147].

βA3/A1-crystallin might represent another therapeutic target in the mTOR pathway, regulating lysosomal degradation. In addition, the overexpression of microRNA-29 in RPE cells enhances autophagic activity as microRNA-29 inhibits the expression of LAMTOR1 (late endosome/lysosomal adaptor, MAPK, and mTOR activator 1)/p18, a key protein located on lysosomal membranes, further inhibiting protein aggregation and enhancing autophagic activity [148] (Figure 4). In addition to the targets related to lysosome activity, S14G-humanin is an important class of cytokine. Humanin members protect RPE cell mitochondria from oxidative damage, and therefore are also therapeutic targets of interest for mitochondrial dysfunction in patients with AMD [149,150].

### 5.3. Advantages of Targeting Autophagy

Compared to phagocytosis, autophagy, as an intracellular waste disposal pathway, has several advantages as a therapeutic target for AMD. First, the autophagy pathway can avoid injuring healthy cells through its self-regulation mechanisms, thereby reducing the risks of adverse side effects and toxicity. Second, the clearance efficiency of the autophagy pathway is higher than that of phagocytosis, allowing for rapid removal of waste materials and maintenance of cellular homeostasis and normal metabolic function. Third, phagocytosis can lead to the production of many inflammatory cells, while the autophagy process is relatively stable and does not cause the release of large amounts of cytokines, thereby avoiding the occurrence of inflammatory reactions [151,152]. Hence, targeting autophagy in RPE is believed to be a more appropriate method for maintaining cell survival and preventing AMD progression. By regulating the initiation and execution of the autophagy pathway in RPE, it is possible to clear aging and damaged organelles and proteins, thus maintaining cellular homeostasis and normal metabolism and potentially providing new therapeutic strategies for preventing and treating AMD.

## 6. Conclusions and Future Outlooks

RPE cells and photoreceptor cells are functional units that closely coordinate to maintain photoreceptor metabolism and renewal. Meanwhile, when RPE cells acquire defects, such as in the MerTk gene, photoreceptor degeneration can occur, severely impacting vision function. Indeed, recent research has elucidated the role of risk factors, such as age, in AMD pathogenesis, revealing that sustained oxidative stress burden can cause cell degeneration, while RPE autophagy protects cells from oxidative stress damage and delays disease progression. Therefore, regulating phagocytosis and autophagy activity to treat AMD has been proposed as a potential new treatment direction.

Currently, mTOR autophagy pathway inhibitors such as rapamycin and AICAR have been implemented for the treatment of various diseases; however, it remains unclear as to whether they can be effectively and safely used to treat AMD. In addition, Beclin 1, as an initiator of the autophagy pathway, and ULK1 participate in the initiation of autophagy and maintain the stability of autophagosomes. In RPE cells, Beclin-1 expression is closely related to the initiation and execution of the autophagy pathway. Moreover, the lncRNA GAS5 plays an important role in cell cycle arrest and apoptosis. In fact, multiple studies have shown that GAS5 is associated with various diseases, including tumors, cardiovascular diseases, autoimmune diseases, etc. Specifically, GAS5 can interact with autophagy-related genes (ATG) and inhibit their expression, inhibiting cell autophagy. Therefore, overexpression of GAS5 may lead to a decrease in autophagy levels in RPE cells, thus affecting their survival and function. Meanwhile, there is currently little research on potential therapeutic targets within the mitochondrial and lysosomal membranes, such as LAMP1/2. Hence, further research should continue to focus on identifying various therapeutic targets and characterizing their mechanisms of action within different AMD phenotypes to inform the development of safe and effective treatment options. In particular, the development of selectively induced or targeted autophagy compounds may provide promising treatment prospects in neurodegenerative eye diseases.

## Figures and Tables

**Figure 1 biomolecules-13-00901-f001:**
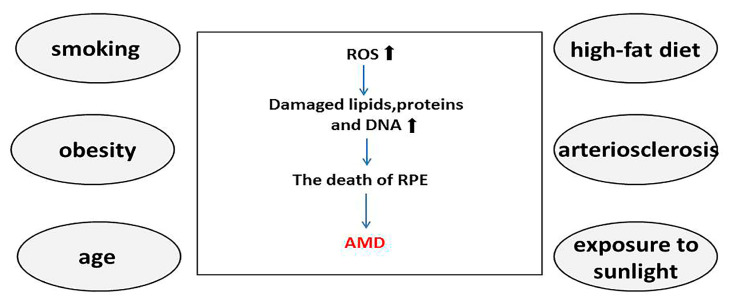
Smoking, obesity, age, high-fat diet, arteriosclerosis, and exposure to sunlight are known major risk factors for age-related macular degeneration (AMD). Reactive oxygen species increase in the presence of these risk factors, inducing oxidative stress and leading to the accumulation of oxidized lipids, proteins, and DNA. This process gradually evolves into degeneration and necrosis of retinal pigment epithelial cells, eventually leading to AMD. The increase of ROS and the accumulation of oxidized lipids, proteins, and DNA are indicated by black arrows, while the progression of AMD is indicated by blue arrows. AMD, age-related macular degeneration; ROS, reactive oxygen species; RPE, retinal pigment epithelium.

**Figure 2 biomolecules-13-00901-f002:**
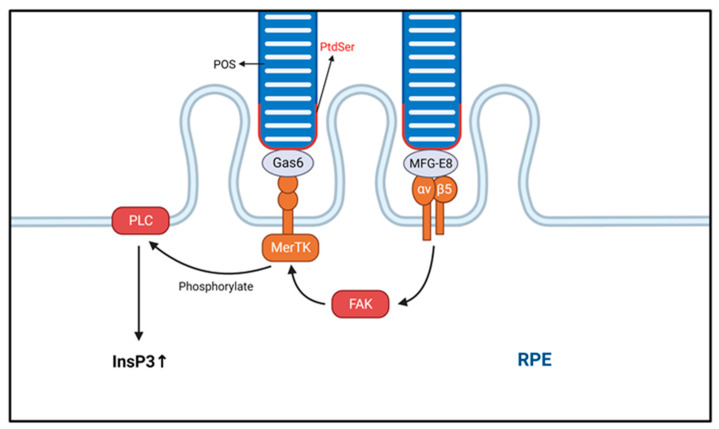
The phagocytic process of the photoreceptor outer segment (POS) is shown. The process of POS ingestion is transduced by integrins. The transduction of intracellular signals begins with the specific binding of macrophage c-mer tyrosine kinase (MerTK). Eventually, the intracellular inositol-1,4,5-trisphosphate (InsP3) content is increased, and InsP3 activation promotes POS ingestion. Integrin αvβ5 binds to phosphatidylserine (PtdSer) via MFG-E8, while milk fat globule-epidermal growth factor (EGF) factor 8 (MerTK) binds via growth arrest-specific protein 6 (Gas6). POS, photoreceptor outer segment; RPE, retinal pigment epithelium.

**Figure 3 biomolecules-13-00901-f003:**
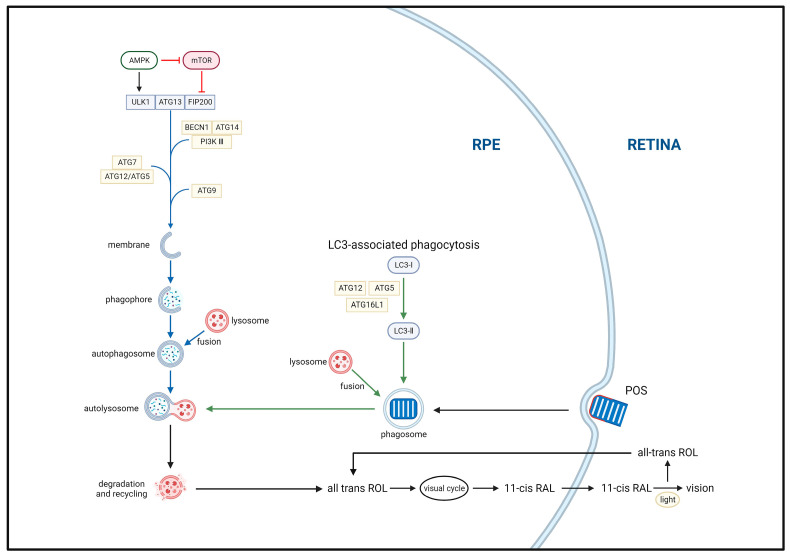
Two main autophagy pathways associated with the classical visual cycle of retinoids. The ULK1, ATG13, and FIP200 complex drives autophagy initiation. Autophagosome formation is primarily mediated by the BECN1, ATG14, and PI3K III complex with double-membrane structures provided by ATG9 vesicles. Autophagosome elongation and closure are coupled through the ATG5 and ATG15 complex. mTOR and AMPK are autophagy inhibitors and inducers, respectively. Autophagy begins with the sequestration of aging, damaged, or unnecessary cellular components, such as organelles and proteins, by double-membrane structures, forming phagophores (autophagosomes). These structures then fuse with lysosomes to form autolysosomes, where the enclosed materials are degraded by acid hydrolases and proteases, and the resulting breakdown products are released into the cytoplasm for recycling. Additionally, LC3-associated phagocytosis (LAP)—a noncanonical form of autophagy—is responsible for the degradation of photoreceptor outer segments (POS) and completion of the retinoid visual cycle. Upon initiation of phagocytosis, LAP immediately recruits ATG12, ATG5, and ATG12L1 to form a complex, as well as LC3 for lipidation. Following POS degradation, all-trans-retinol is converted to 11-cis-retinol in the RPE and transported out of the cell, completing the classical visual cycle of retinoids.

**Figure 4 biomolecules-13-00901-f004:**
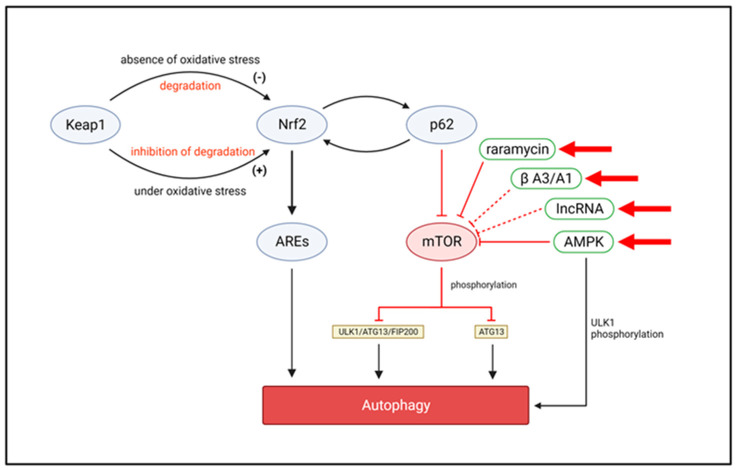
Autophagy pathways in age−related macular degeneration. Possible therapeutic targets are indicated by arrows.

## Data Availability

Not applicable.

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
