# Peer review of "The Role of Retinal Pigment Epithelial Cells in Age-Related Macular Degeneration: Phagocytosis and Autophagy"

_biomolecules, 2023, doi:10.3390/biom13060901_

Round 1

Reviewer 1 Report

The article titled Retinal Pigment Epithelial Cells in Age-Related Macular Degeneration: Phagocytosis and Autophagy is a good review of the role of the RPE in regulating the retina in general and specifically in age-related macular degeneration. The explanation of the physiological and pathophysiological processes of phagocytosis and autophagy are very interesting and are expressed in a simple, understandable way and with extensive bibliographic support. The figures, however, especially the second one, are very simple and do not give value to such rigorous explanations of the statements. It would be recommended to adapt the figures. The rest of the article is of high scientific quality.

Author Response

We sincerely appreciate the valuable comment.

We have replaced Figure 2 and Figure 4.In addition to that, we added 3.2.1 Autophagy Mechanism and Figure 3(on page 5),and in this part, the mechanism of autophagy is introduced.

Reviewer 2 Report

The review paper covers basic information about oxidative stress and mitochondria dysfunction in RPE cells in AMD, focusing on heterophagy and autophagy. It is a well-written review article, the arguments are cohesive, and the structure is easy to follow. However, it needs more novelty of insights.    Here are some suggestions
  1. The authors focus more on the autophagy of RPE cells towards the end of the article. The authors can provide novel insights based on published original work on why this is a better approach than targeting heterophagy.
  2. There have been a few recent review articles on the same topic. For example, Kaarniranta et al 2020 PMID 32298788, Tong et al 2022 PMID 35912040 and Yako et al 2021 PMID 34688622. It would be beneficial to the readers if the authors could point out the further or novel insights that the paper will provide in the introduction. 
  3. The authors could include a section on the latest developments or therapeutic targets in the field. 
  Minor comment

Line 120. The authors suddenly used the word heterophagy to replace phagocytosis. If they are the same process, it should be mentioned.

Author Response

Comment 1 from revewer:The authors focus more on the autophagy of RPE cells towards the end of the article. The authors can provide novel insights based on published original work on why this is a better approach than targeting heterophagy.

Our reply:We feel great thanks for your professional review work on our paper.As you are concerned, in the fifth section, we have made a lot of modifcations in terms of content(on page 10-13,and revised portion are marked in red in the paper).In this section, potential Therapeutic Targets for AMD has been significantly modified. Also, the advantages of phagocytosis targets, autophagy targets, and targeted autophagy have been introduced.

Comment 2:There have been a few recent review articles on the same topic. For example, Kaarniranta et al 2020 PMID 32298788, Tong et al 2022 PMID 35912040 and Yako et al 2021 PMID 34688622. It would be beneficial to the readers if the authors could point out the further or novel insights that the paper will provide in the introduction.

Our reply:We sincerely appreciate the valuable comment.Within the introduction, new insights regarding the therapeutic targets of phagocytosis and autophagy in the RPE have been added(on page 1-2,and revised portion are marked in red in the paper).In addition, we have made corresponding additions in the abstract of the review.

Comment 3:The authors could include a section on the latest developments or therapeutic targets in the field.

Our reply:We think tihis is an excellent suggestion.In the sixth section, Conclusions and Future Outlooks, we have re-written this section to discuss the latest progress and prospects of therapeutic targets(on page 12-13,and revised portion are marked in red in the paper).

Comment 4:Minor comment Line 120. The authors suddenly used the word heterophagy to replace phagocytosis. If they are the same process, it should be mentioned.

Our reply:We sincerely thank the reviewer for careful reading.As suggested by the reviewer,we have corrected the “heterophagy” into “phagocytosis”.